# Sand flies and Toscana virus: Intra-vector infection dynamics and impact on *Phlebotomus perniciosus* life-history traits

Lison Laroche[1,2]*, Anne-Laure Bañuls[1], Rémi Charrel[3], Albin Fontaine[3,4], Nazli Ayhan[3,5], Jorian Prudhomme[1,6]

1 MIVEGEC, Université de Montpellier – IRD – CNRS, Centre IRD, Montpellier, France, 2 Department of Life Sciences, Imperial College London, London, United Kingdom, 3 Unité des Virus Emergents (UVE: Aix-Marseille Univ, Universita di Corsica, IRD 190, Inserm 1207, IRBA), Marseille, France, 4 Unité de virologie, Département Microbiologie et maladies infectieuses, Institut de Recherche Biomédicale des Armées (IRBA), Marseille, France, 5 Centre National de Référence des Arbovirus, Marseille, France, 6 Université de Rennes, Inserm, EHESP, IRSET (Institut de Recherche en Santé Environnement Travail), UMR_S 1085, Rennes, France

☉ These authors contributed equally to this work.

* lisonlaroche14@gmail.com

## Abstract

Toscana virus (TOSV) is a leading cause of summer viral meningitis in Southern Europe (Central Italy, south of France, Spain and Portugal) and can cause severe neurological cases. Within the Mediterranean basin, it is transmitted by hematophagous sand flies belonging to the *Phlebotomus* genus. Despite the identification of the primary TOSV vectors, the viral developmental cycle in vector species remains largely unknown. Limited research has been conducted on transmission dynamics and the vector competence and vectorial capacity of the principal TOSV vector, *Phlebotomus perniciosus*. In this context, we investigated the intra-vector TOSV infection dynamics in *Ph. perniciosus*, as well as its impact on the vector life history traits. Female sand flies were experimentally infected with TOSV through an artificial blood meal. Systemic dissemination of the virus was observed approximately three days post-infection, potentially resulting in a short extrinsic incubation period. Moreover, the study revealed a longer hatching time for eggs laid by infected females. This research brought additional experimental insights regarding the vector competence of *Ph. perniciosus* but also provided the first insight into TOSV developmental cycle and its impact on the vector. These findings prompt further exploration of TOSV transmission dynamics, raise new hypotheses on the virus transmission and highlight the importance of follow-up studies.

## Author summary

Toscana virus (TOSV) is a reemerging sandfly-borne virus causing neuroinvasive infections in humans. This virus is endemic in the Mediterranean basin, with a potential risk of introduction in northern Europe and Asia. Despite decades of research, few studies have

**Data Availability Statement:** All relevant data are within the manuscript and its Supporting information files.

**Funding:** This research was funded by the IRD (Institut de Recherche pour le Développement) to LL, CNRS (Centre National de la Recherche Scientifique) to LL, UM (Université de Montpellier) to LL and INFRAVEC2 (https://infravec2.eu/, grant agreement N° 731060 2017-2022) to LL. L.L. was financially supported by an UM doctoral fellowship and obtained a grant Key Initiative Montpellier: Risks and Vectors (KIM RIVE), supported by Montpellier University of Excellence (MUSE) and défi clé RIVOC, supported by Région Occitanie), with J.P. as PI, which fund part of the material for this study. This work was also supported by the European Commission (European Virus Archive Global project (EVA GLOBAL, grant agreement No 871029) of the Horizon 2020 research and innovation programme (european-virus-archive. com/). LL received salary from University of Montpellier and J.P received salary from INFRAVEC2 project. The funders had no role in study design, data collection and analysis, decision to publish, or preparation of the manuscript.

**Competing interests:** The authors have declared that no competing interests exist.

focused on the development cycle of TOSV in sand flies and the dynamics of transmission. Here, we provide a comprehensive study of the intra-vector dynamics of TOSV infection and its impact on both vector biology and consequently on transmission. Through experimental infections of the major vector *Phlebotomus perniciosus*, we not only brought additional experimental insights regarding vector competence but also provided the first insight into the TOSV developmental cycle in the vector by estimating the extrinsic incubation period at six days. Our study reveals an impact of TOSV infection on vector egg hatching time that could lead to a delayed emergence of infected sand flies, with a potential impact on transmission. Our findings encourage further exploration of transmission dynamics, raise new hypotheses on alternative transmission pathways, and emphasize the importance of follow-up studies.

## Introduction

Toscana virus (TOSV) is an enveloped, protein-encapsidated, tri-segmented RNA virus that belongs to the *Phlebovirus toscanaense* species, *Phlebovirus* genus, *Phenuiviridae* family and Bunyavirales order [1]. Initially isolated in 1971 from *Phlebotomus perniciosus* and *Phlebotomus perfiliewi* in central Italy, TOSV is endemic in the Mediterranean basin where at least 250 million people are at risk of infection [2,3]. Despite the increasing frequency of TOSV infections, the virus remains largely neglected due to inadequate diagnostic tools [4]. As typical with arthropod-borne viruses (arboviruses), many TOSV infections are unreported since they are either asymptomatic or cause only mild symptoms [5]. This virus affects individuals across various age groups, with a median age of 44.5 years (under review). Toscana virus has a particular neurotropism and is one of the primary causes of meningitis and encephalitis in endemic regions [6]. A total of 73.4% of patients suffering from meningitis and encephalitis present at least one of neurological manifestation due to TOSV infection (under review). Human infections occur during warm seasons, with a peak in the hottest months in relation to vector activity [3]. Three genetic lineages (A, B and C) of TOSV circulate within the Mediterranean area, but no differences in pathogenicity have been demonstrated to date [7,8]. Toscana virus is transmitted to humans through a bite of an infected female sand fly [9] and its geographical distribution is related to competent sand fly vector presence and continues to expand. Currently, four sand fly species established in the Mediterranean area are recognized or suspected as TOSV vectors: *Ph. perniciosus*, *Ph. perfiliewi*, *Phlebotomus sergenti* and *Phlebotomus neglectus* [7]. However, other sand fly species may also contribute to the virus transmission, such as *Phlebotomus tobbi*, *Phlebotomus longicuspis* and *Sergentomyia minuta* [7].

Due to its neuroinvasive nature, TOSV has emerged as the most significant sandfly-borne phlebovirus for public health, warranting further investigation into its natural cycle [10]. However, to date, limited information is available on natural cycle and transmission of TOSV particularly with regard to the phlebovirus infection dynamics (the progression of the virus within a host organism over time) in the vectors or in animals or humans [11–13]. While transovarial and venereal transmissions have been experimentally observed in its primary vector, *Ph. perniciosus* [14], their significance in the natural TOSV cycle remains uncertain and warrants further investigation. Additionally, TOSV has been isolated in Italy and Morocco from naturally infected males of *Ph. perniciosus*, however males are not haematophagous and these results has been interpreted as evidence of transovarial and/or venereal transmission under natural conditions [15,16]. However, recently it was hypothesized that transmission from infected to uninfected sand flies during sugar feeding might play a significant role in the virus natural cycle

[17]. This mode of transmission has already been shown experimentally with Massilia virus, another phlebovirus genetically related but distinct from TOSV [18]. Moreover, TOSV persistence and infectivity in sand fly sugar meal has recently been demonstrated in laboratory conditions [17]. Nevertheless, the relevance of this alternative transmission route to TOSV cycle in natural ecosystems remains to be confirmed.

In recent decades, arboviruses have emerged as significant causes of death and disability worldwide, highlighting the need for a better understanding of their transmission dynamics [19–21]. Most of this research has concerned mosquito-borne viruses compared to studies on sandfly-borne viruses, reflecting the historically higher impact of mosquito-transmitted diseases on public health. Arbovirus epidemics are influenced by both intrinsic factors (vector competence, virus strain or genetic lineage, virus dose-effect, etc.) and extrinsic factors (temperature, rainfall, human activity, etc.) that can influence vector biology [22–24]. These factors can affect the infection and transmission dynamics of arboviruses [25]. Moreover, certain studies have shown the effects of arbovirus infection on mosquito life-history traits, such as survival and fecundity, with a potential impact on transmission dynamics [26–28]. Therefore, to comprehensively address the public health impact of sandfly-borne viruses, such as TOSV, further studies are needed to examine their infection dynamics.

In our study, we aim to understand the impact of TOSV infection on biology of its primary vector *Ph. perniciosus* and its potential effect on transmission. We set up experimental infections to achieve two main objectives. The first one is to determine the infection dynamics of TOSV in *Ph. perniciosus* by confirming the vector competence for TOSV lineage B and explore how different TOSV dose influence the kinetics of infection within its vector. The second objective is to examine the impacts of TOSV infection on the vector life history traits, including survival and fecundity, and investigate the potential effects on transmission dynamics. Ultimately, this work will provide a deeper understanding of TOSV transmission mechanisms and the impact of infection on vector biology.

## Methods

### 1. Sand fly rearing

In order to realize experimental studies, we established (IRD Vectopôle, Montpellier, France) a *Ph. perniciosus* (from Pr. Ricardo Molina, Laboratorio de Entomología Médica, Instituto de Salud Carlos III, Madrid, Spain; 'Boadilla *Ph. perniciosus*' strain collected in Madrid, Spain, in 1987) colony according to previously described protocols [29–33] with few modifications [34]. We chose this species because it is the major vector common to TOSV and *Leishmania infantum* in the Mediterranean basin. Sand fly colonies were maintained in climate chamber under standard conditions (26 ± 1˚C, 80% relative humidity (rh), 14h: 10h light: dark cycle). Adults were fed with cotton pads soaked in 50% organic sugar solution. Blood meal were realized once a week with glass feeders filled with rabbit blood maintained at 37˚C and covered with a 3-day-old chick skin membrane. Feeding was performed for six hours with a regular manual blood mix to avoid clotting. After 24h, blood-fed females were transferred to plastic pots filled with a 1cm layer of plaster of Paris. These pots were stored in plastic boxes containing moistened sand. After hatching, larvae were fed with a macerated mixture of 40% rabbit excrement, 40% rabbit food and 20% mouse food. Newly emerged adults were transferred to rearing cages or in small cages for experimental infection.

### 2. Toscana virus experiments

**Virus isolates.** Lyophilized TOSV aliquots (strain MRS2010, lineage B, Genbank ID: KC776214 KC776215 KC776216) were provided by the laboratory UVE (Unité des Virus

Emergents, Marseille, France). Vero E6 cells were grown in monolayers in Minimum Essential Medium (MEM, Gibco) complemented with 7% heat-inactivated Fetal Bovin Serum (FBS, Eurobio Scientific), 1% L-glutamine (Gibco) and 1% penicillin-streptomycin (Gibco) at 37°C with 5% $CO_2$. Stocks of TOSV were obtained by dissolving the lyophilisates in pure water, and used for infecting Vero E6 cells at 0.1 multiplicity of infection (MOI). After five days of post infection, supernatant was collected. TOSV stocks at a concentration of $4.2 \times 10^6$ 50% tissue culture infective dose ($TCID_{50}$/ml), at passage 2, were aliquoted and stored at -80°C.

**Experimental infection.** Experimental infections of females *Ph. perniciosus* with TOSV were performed in enhanced biosafety level 3 (BSL-3) laboratory (IRD Vectopôle, Montpellier, France). Experimental infections were realized as previously described for *Leishmania* infection [35] modified for TOSV. Twenty-four hours prior to the experiment, 5 to 9-day-old females from the colony were starved by deprivation of sugar meal. Before infection, approximately 100 females were transferred into a feeding pot and stored in the BSL-3 climate chamber under standard conditions (26 ± 1°C, 80% rh) to acclimatize. Approximately 10% males were added to mate and stimulate females to take blood [29]. Glass feeders were fill with heat-inactivated rabbit blood and covered with a chick skin membrane. For infected batches, TOSV (2nd passage of viral culture) was added at a final concentration of $10^2$ $TCID_{50}$/ml (dose 1), $10^4$ $TCID_{50}$/ml (dose 2) or $10^6$ $TCID_{50}$/ml (dose 3). For control batch (control), the same volume of MEM culture medium, without virus, was added. The complete system was set up in the climate chamber where the feeders were connected to a 37°C circulating water bath and fixed to the feeding pots (Fig 1). Feeding was performed for six hours, with reduced brightness, and regular manual blood mix to avoid clotting. Six hours of blood feeding was chosen as TOSV remains infectious and at the same amount in the blood meal under these conditions as previously described [17]. As engorged females are fragile, it is recommended to handle them after the first 24 hours of digestion [29]. Thus, they were sorted on ice the day after blood meal. Males and non-engorged females were killed and removed. All the experiments were replicated twice.

a. Infection dynamics

Engorged females were placed by 30 in cardboard boxes in climate chamber, under standard conditions, with a daily intake of cotton pads soaked in 50% organic sugar solution.

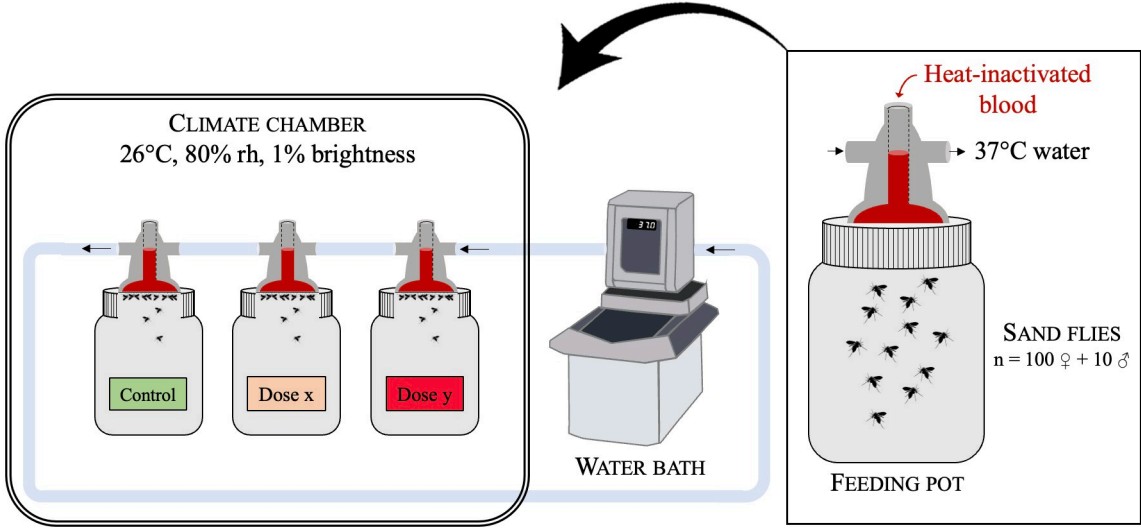

**Fig 1. Blood feeding system in the BSL-3 laboratory.** rh: relative humidity.

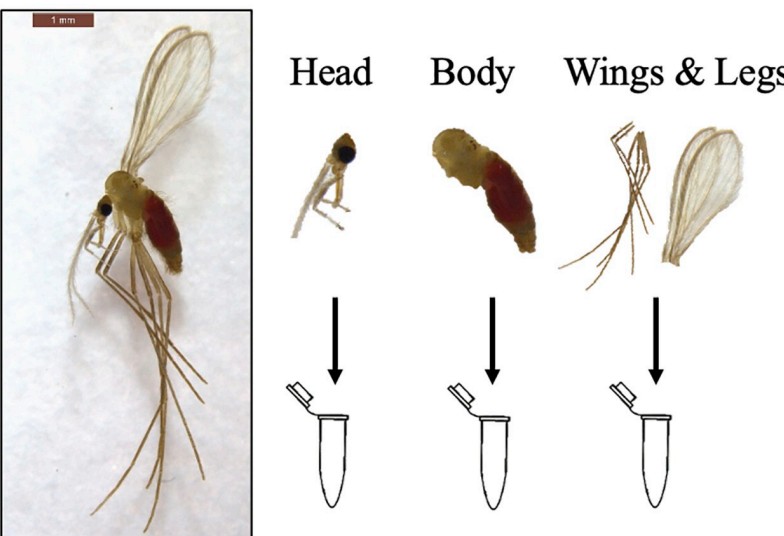

**Fig 2. Dissection of infected females in three parts: Head, body, wings and legs for Toscana virus detection (Photo by J. Prudhomme).**

Three groups were set up for this experiment, infected with dose 1, dose 2 and dose 3 of TOSV. Between 10 to 30 individuals were dissected in three parts: head, body (thorax and abdomen), wings and legs, every two days for 15 days (Fig 2), and stored at -80°C for further TOSV detection and quantification analysis. In order to detect virus released by the females, the sugar cotton pads were collected daily, and stored at -80°C. One cotton per day at two, four, six, eight and 10 days post infection (dpi) with dose 3 only (as a proof of concept, to ensure the viral detection) were analyzed with the same RT-qPCR protocol as the sand fly samples.

b. Impact of infection on life-history traits

To investigate the potential impact of TOSV infection on sand fly life-history traits, vector survival and fecundity traits, which are two important traits in the virus transmission, were measured. For this experiment, engorged females were individually dispatched in small egg-laying pots filled with a 1cm layer of plaster of Paris, pooled by group in plastic boxes containing moistened sand and placed in climate chamber under standard conditions [33]. A pool of sand flies was set aside in a cardboard box for TOSV infection verification by RT-qPCR. Groups were set up for this experiment, control with uninfected individuals, and two tests with individuals infected by different TOSV doses ($10^4$ and $10^6$ $TCID_{50}$/ml). Life history traits were measured daily and individually for each female: death date, oviposition date, eggs laid number, egg-hatching date and larvae number. Immature stages were only monitored up to first instar larva, due to potential cannibalism between larvae [36], and larvae were killed directly after hatching.

To measure infection impact on female survival, the experiment was performed with a different batch of females. After blood feeding, engorged females were transferred to cardboard boxes with a maximum of 30 individuals per box in climate chamber and maintained under standard conditions (26 ± 1°C, 80% relative humidity (rh), 14h: 10h light: dark cycle). Five blood-fed females were killed, dissected and stored at -80°C for RT-qPCR analysis directly after oral infection to measure the initial amount of virus ingested. The survival time of two

female groups, fed with (i) healthy blood (control) and (ii) infected blood with the highest dose of TOSV (dose3), was measured every day until the death of all individuals. In order to check the potential impact on survival, this experiment was only realized with the highest virus dose. To prevent the fungi development which could affect the survival of the sand flies, dead females were removed and the sugar source was changed every day.

**Virus detection.** Viral load was analyzed by RT-qPCR. Entire female or each part (head, body, wings and legs) were individually crushed in 300μl of MEM (enriched with 1% penicillin–streptomycin, 1% (200mM) l-glutamine, 1% kanamycin, and 3% amphotericin B (Gibco)). Sand fly tissues were homogenized by a Tissuelyser II (Qiagen) using 3mm tungsten carbide beads (Qiagen), and centrifuged at 10000 rpm for 5min. A volume of 200μl homogenate supernatant was used for nucleic acid extraction with the QIAcube (Qiagen) machine, using Virus Extraction Mini Kit (Qiagen). The RT-qPCR assays were performed with a SuperScript III Platinum One-Step RT-qPCR Kit with ROX (Invitrogen—Thermo Fisher Scientific) on a QuantStudio 12K Flex thermocycler (ThermoFisher). A volume of 5μl of RNA was added to 20μl of mix containing 12.5μl of 2X Reaction Mix, 0.5μl of Superscript III RT/Platinum Taq Mix, and primers and probes STOS at 10μM [37]. Negative (pure water) and positive controls (at $4.81 \times 10^4$ RNA in vitro transcribed copies/ml as described by Beckert & Masquida [38]) were included in each RT-qPCR run. A standard curve was carried out for absolute quantification of TOSV RNA through RT-qPCR (S1 Fig). This standardised RT-qPCR protocol was used to identify the number of viral particles per body part, assuming that the number of cells per body part is consistent from one sand fly to another.

TOSV titers were determined by end-point dilution assay. Tenfold dilutions were used to infect confluent Vero E6 cells in a 96-well plate in MEM (5% FBS, 1% penicillin-streptomycin, 1% kanamycin, 3% amphotericin B (Gibco)) at 37°C and 5% $CO_2$. Wells were classified as positive (cytopathic effect) *versus* negative (no cytopathic effect) at five dpi and $TCID_{50}$/ml was calculated according to the Reed-Muench method [39].

## 3. Statistical analysis

Data analysis were carried out with R through Rstudio integrated development environment version 2022.07.1–2009–2022 [40]. Survival was examined with the Kaplan Meier product limit estimator, and a log-rank test was used to compare survival distributions between uninfected and TOSV-infected sand flies, using the packages survival [41] and survminer [42]. Egg hatching times were also examined with the Kaplan Meier method, and a log-rank test was used to compare hatching time distributions between uninfected and TOSV-infected sand flies with both doses ($10^4$ and $10^6$ $TCID_{50}$/ml). The significance of the TOSV doses ($10^4$ and $10^6$ $TCID_{50}$/ml) and the effect of time post-infection on the infection dynamics was tested by general linear mixed models (GLM). Systemic infection dynamics process is linked to the extrinsic incubation period, which is the time required for an arthropod to progress from being infected to becoming infectious. Probabilities of systemic infection (virus dissemination from sand fly midgut to secondary tissues) according to the TOSV dose or/and time post-infection were estimated with a logistic model. Median systemic infection doses were calculated based on logistic regression parameter estimates.

## Results

### Toscana virus infection dynamics

A total of 1836 female sand flies were used to determine the infection dynamics of TOSV in *Ph. perniciosus*. The number of sand flies used in each group depended on the productivity of the sand fly colony at the beginning of the experiment. Respectively, 73 (20%), 761 (68%) and

**Table 1. Number of *Phlebotomus perniciosus* dissected per day to determine Toscana virus infected individuals by Toscana virus specific RT-qPCR, depending on viral loads.**

|  | Days post infection | | | | | | |
|---|---|---|---|---|---|---|---|
|  | 3 | 5 | 7 | 9 | 11 | 13 | 15 |
| Dose 1 ($10^2$ TCID$_{50}$/ml) | 10 | 5 | 5 | 6 | 5 | 5 | 2 |
| Dose 2 ($10^4$ TCID$_{50}$/ml) | 20 | 47 | 39 | 46 | 52 | 55 | 45 |
| Dose 3 ($10^6$ TCID$_{50}$/ml) | 20 | 20 | 20 | 20 | 10 | 10 | 4 |

335 (77%) of females fed on TOSV infected blood with dose 1 ($10^2$ TCID$_{50}$/ml), dose 2 ($10^4$ TCID$_{50}$/ml) and dose 3 ($10^6$ TCID$_{50}$/ml). After blood feeding, sand flies were dissected every two days, however due to the daily natural mortality (10–20%), it was not possible to dissect all blood-fed females. The dissected number of females are summarized in Table 1.

After testing sand flies: head, body, wings and legs, at dose 1, TOSV was detected by RT-qPCR in the bodies of two out of 10 females at three dpi and one out of five females at seven dpi. No virus was found in heads or wings and legs after infection with dose 1. For the dose 2 infection, we observed viral load higher than $10^4$ RNA copies/ml at three dpi in female bodies. This viral load increased then to $10^6$ RNA copies/ml at 11 dpi, indicating viral replication (Fig 3A). The viral load in the heads, wings and legs was lower compared to bodies, but started to increase from four to five dpi. We observed a stabilization of the viral load in the wings and legs from nine dpi, remaining between $10^2$ and $10^3$ RNA copies/ml until 15 dpi. For the infection with dose 3, we detected TOSV RNA in all sand fly parts (head, body, wings and legs) until 15 dpi (Fig 3B). The viral load increased by almost one log from three to 13 dpi in bodies. The viral load started to increase in the head, wings and legs from four to five dpi, but with a lower RNA copy number. The analysis of significance of the TOSV dose on post-infection day showed that the viral load has a significant effect on the infection dynamics in sand fly bodies (GLM, $P = 0.0002$) (data from dose 1 infection was not included in the analysis due to low sample size). The incidence of infection exhibited a dose-dependent augmentation in response to TOSV exposure, with a significant effect of post-infection time (GLM, $P = 0.0314$).

Systemic infection rates were measured by comparing the number of sand flies with infected heads to infected bodies. The within-host dynamics of systemic TOSV infection were represented for the two viral doses by fitting a non-linear regression model (Fig 4). The rates of systemic infection did not saturate at 100% for either TOSV dose. The estimated time of 50% systemic infection was approximately five (SE: 0.1) days for sand flies infected with dose 2. For sand flies infected with dose 3, the estimated time of 50% systemic infection was three (SE: 0.3) days.

To determine the virus infectivity in sand fly samples, TOSV titers of bodies, heads, wings and legs were determined by an end-point dilution assay (TCID$_{50}$). Four sand flies per day (every two days) were analyzed by titration for the highest infection (dose 3). Of the 84 samples analyzed (n = four sand flies * three parts * seven days = 84), seven sand fly bodies were found positive: at 1, 3, 5, 7, 9, 11 and 14 dpi (Table 2). For each dpi, 1 body out of 4 sand fly samples was positive, representing a 25% infection rate. There were no positive samples for heads, legs and wings. Only the highest dose 3 was analyzed by titration for this experiment to ensure results were obtained, but it would also be necessary to further analyze samples from dose 2.

Of the sugar cotton pads analyzed at two, four, six, eight and 10 dpi, in order to detect the virus released by the females during sugar feeding, TOSV RNA was detected at two, six and eight dpi. According to the standard curve, the virus limit detection was $10^{-2}$ (log10) RNA copies/ml. An average of $10^3$ (log10) RNA copies/ml of TOSV was detected in samples

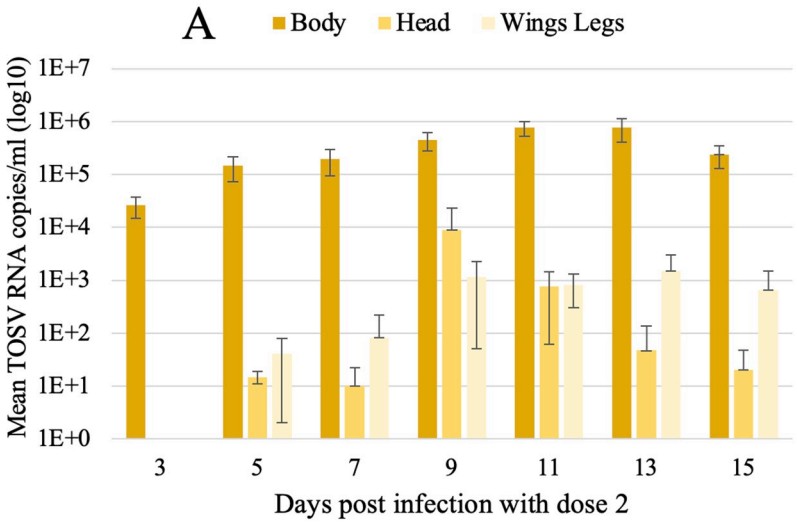

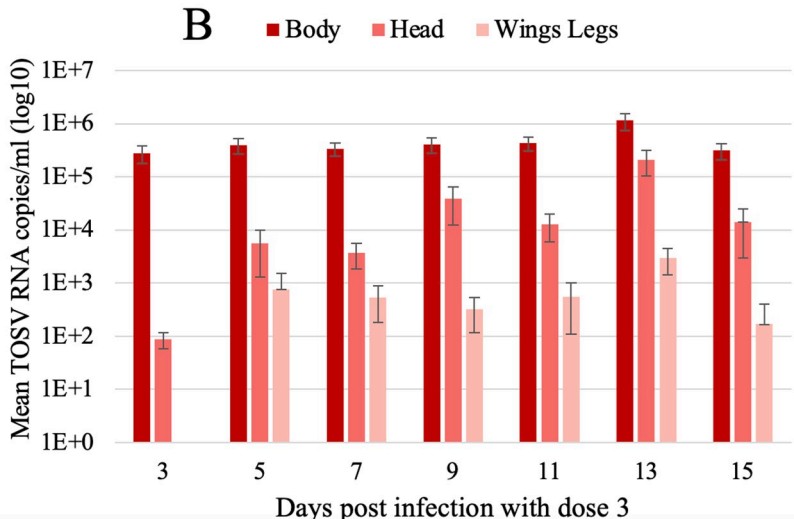

**Fig 3. Mean log10 viral particles in body, head, wings and legs of infected female sand flies fed with dose 2 (A) and dose 3 (B) of Toscana virus over time post infection.** Toscana virus RNA levels in the sand fly parts were quantified by RT-qPCR. The error bars correspond to the standard deviation.

collected at two dpi (30 female pool). At six (27 female pool) and eight (20 female pool) dpi, TOSV RNA was detected in the sugar samples at an average of $10^3$ and $10^4$ (log10) RNA copies/ml, respectively.

## Toscana virus impact on vector life history traits

To investigate the impact of TOSV infection on *Ph. perniciosus* survival life-history trait, 140 and 98 sand flies were used for the control (uninfected females) and the test (infected females with dose 3), respectively. Blood-fed females were euthanized and dissected directly after oral infection. They ingested an average of $2.5 \times 10^5$ RNA copies/ml. Using the Kaplan-Meier method, the survival distributions were not significantly different between the two groups (log-rank test, $P = 0.92$). The survival probabilities were not statistically different for

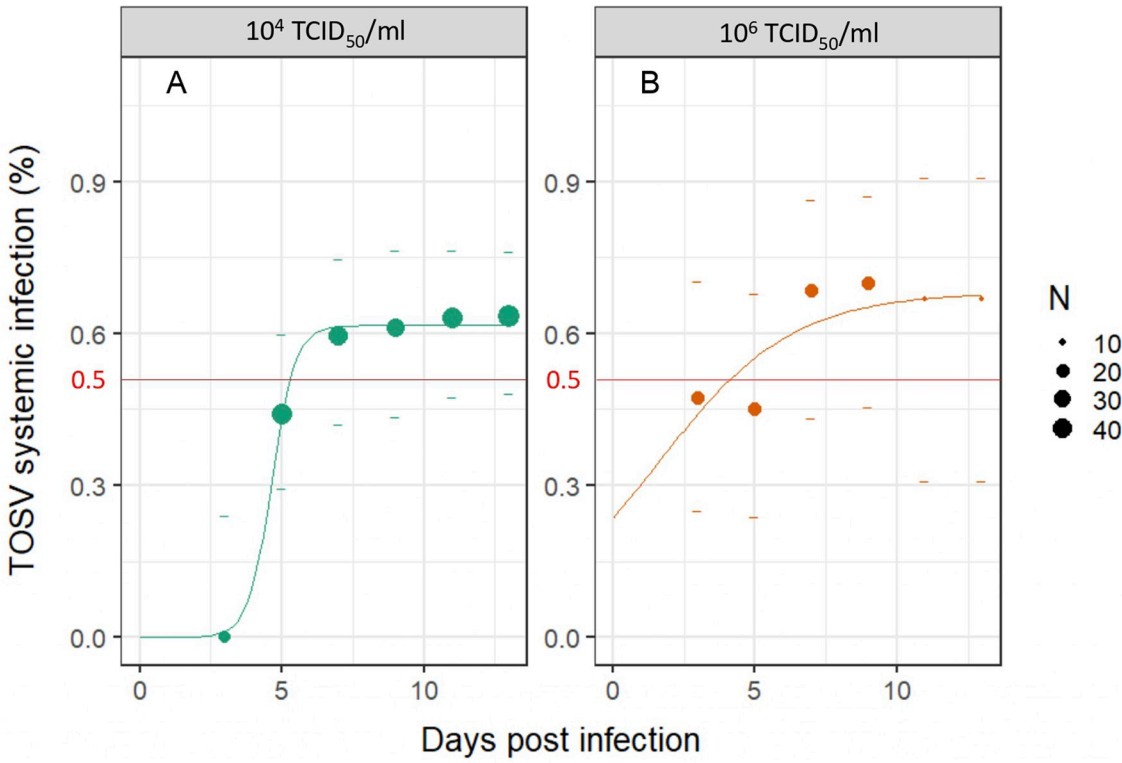

**Fig 4. Systemic infection dynamics for *Phlebotomus perniciosus* infected with $10^4$ TCID$_{50}$/ml (A) and $10^6$ TCID$_{50}$/ml (B).**
Cumulative rates of systemic (disseminated) infections over time post Toscana virus infection are represented as point, dot size indicates the number of samples. Dashes represent the 95% confidence intervals of rates, and the red line represents 50% systemic infection. The fitted values obtained with a non-linear regression model are represented by a line. Fig 4 was made using the package ggplot2 [43].

uninfected and infected sand flies (S2 Fig). The last infected female was purposely euthanized at 45 dpi, dissected and analyzed by TOSV RT-qPCR. An amount of 2.2x10$^4$ RNA copies/ml was found in the body of this sample at 45 dpi. However, this data was considered censored for the survival analysis.

For the experimental investigation assessing the infection impact on vector fecundity life-history traits (egg numbers, oviposition time, larvae numbers and hatching time and rate), a cohort of 543 female specimens was used. The blood-feeding rates were relatively low for each group (50%). A total of 85 (control), 94 (dose 2) and 96 (dose 3) individuals were placed in individual laying pots. The different life-history trait data were collected during 35 days of monitoring of control and TOSV infected sand flies until the last hatching (Table 3). Among all the parameters studied, only hatching time of test groups dose 2 (GLM test, $P = 0.0247$) and dose 3 (GLM test, $P = 5.61e-07$) were significantly different from the control group (Fig 5 and S1 Table). We have confirmed these significant results with the Kaplan-Meier method (S3

**Table 2. Toscana virus titers (TCID$_{50}$/ml) in *Phlebotomus perniciosus* bodies infected with dose 3 ($10^6$ TCID$_{50}$/ml), per day post infection (dpi).**

| Dpi | 1 | 3 | 5 | 7 | 9 | 11 | 14 |
|---|---|---|---|---|---|---|---|
| TCID$_{50}$/ml | 9.3 | 9.3 | 29 | 20 | 20 | 9.3 | 9280 |

**Table 3. Summary of the experiments with the different life history traits of *Phlebotomus perniciosus* studied.** The control group represents uninfected blood fed sand flies, and the tested groups Dose 2 and Dose 3 represent blood fed infected sand flies with $10^4$ TCID$_{50}$/ml and $10^6$ TCID$_{50}$/ml, respectively. BF: Blood-Fed, BF*: Blood-Feeding, No: Number of. Means are indicated with their standard deviation.

| | No. ♀ initial | No. ♀ BF | BF* rate (%) | Survival time (days) | No. eggs | Oviposition time (days) | No. larvae | Hatching time (days) | Hatching rate (%) |
|---|---|---|---|---|---|---|---|---|---|
| Control | 170 | 85 | 50 | 7.6 ± 0.4 | 29.0 ± 2.2 | 6.0 ± 0.4 | 21.8 ± 2.0 | 5.9 ± 0.1 | 70 |
| Dose 2 | 182 | 94 | 52 | 9.2 ± 0.3 | 32.2 ± 2.0 | 7.9 ± 0.4 | 21.1 ± 1.8 | 6.4 ± 0.2 | 60 |
| Dose 3 | 191 | 96 | 50 | 8.6 ± 0.4 | 27.9 ± 2.1 | 7.2 ± 0.3 | 19.5 ± 1.6 | 7.1 ± 0.1 | 65 |

Fig). Hatching time shows a positive correlation with the quantity of TOSV ingested (Table 3 and Fig 5).

## Discussion

Toscana virus is an emerging and neglected sandfly-borne phlebovirus in the Mediterranean basin. This virus is one of the most frequent cause of summer viral meningitis in geographic areas where human population are exposed and can lead to severe neurological cases [6]. Despite a significant effect on human health, limited research has been conducted on its

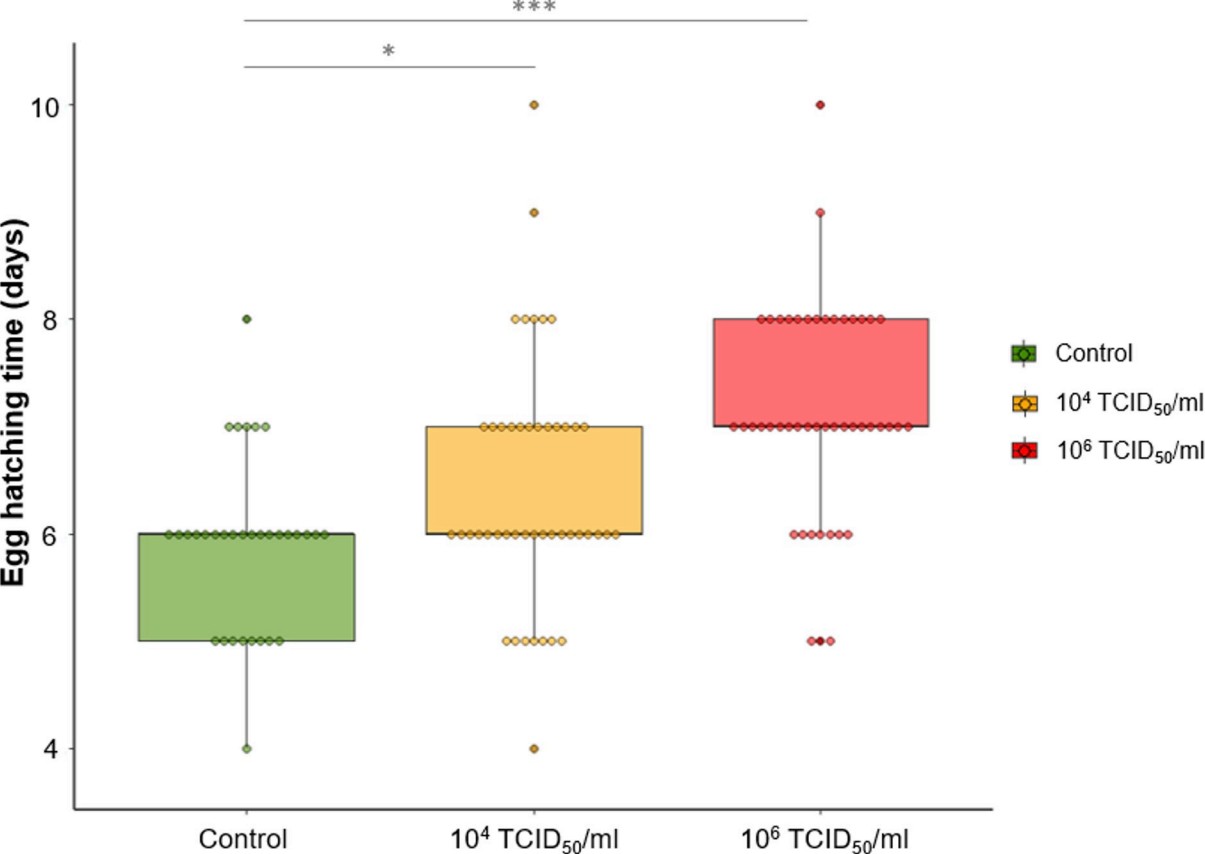

**Fig 5. Boxplot representing the impact of Toscana virus infection on *Phlebotomus perniciosus* egg hatching time.** Dots represent the average hatching time of eggs from one female. In green, the control group of uninfected females; in yellow, the test group of females infected with dose 2 ($10^4$ TCID$_{50}$/ml); in red, the test group of females infected with dose 3 ($10^6$ TCID$_{50}$/ml). The stars are the result of the GLM-based comparison (p-value significance: 0.0001 '***' 0.001 '**' 0.01 '*' 0.05).

transmission dynamics. Although the main sand fly vectors of TOSV have been identified, the viral development cycle in its vectors remains largely unexplored [44]. While numerous field studies have detected TOSV in *Ph. perniciosus* populations, only a few experimental studies have yet assessed the vector competence and vectorial capacity of this species [13,45,46].

In our study, the first objective was to investigate experimentally infection dynamics in the vector using three different viral doses. In general, vector competence (intrinsic ability of a vector to be infected and further transmit a virus) is a progressive process that begins when the vector takes an infectious blood meal from a viremic host. After initial infection and replication in the midgut, the virus spreads from the midgut to secondary tissues like haemocoel (systemic infection), then to the salivary glands (located in the thorax of sand flies), where it may be released in saliva and transmitted during a subsequent blood meal [47]. Herein, we found viral RNA in head and thorax [48], until at least 15 dpi, as well as in wings and legs (Fig 3). In addition, the viral titration in sand fly bodies infected with dose 3 ($10^6$ $TCID_{50}$/ml) showed that the TOSV was still infectious until 14 dpi (Table 2). In this study, we confirmed the TOSV replication and dissemination within *Ph. perniciosus*.

Additionally, our study demonstrated a direct correlation between the infection rates in sand flies and the infectious dose in the ingested blood meal (Fig 3). When sand flies were infected with a low TOSV dose ($10^2$ $TCID_{50}$/ml), virus particles did not disseminate in the vector, and only a few body samples contained TOSV before sand fly defecation (blood meal digestion and excretion). Therefore, it can be assumed that ingestion of low virus dose does not lead to systemic infection. In contrast, in sand flies having ingested $10^4$ $TCID_{50}$/ml and $10^6$ $TCID_{50}$/ml, TOSV disseminated into the secondary tissues as showed by detection of TOSV RNA for at least 15 days (Fig 4). Based on our experimental findings, the minimum infectious dose for TOSV systemic infection in *Ph. perniciosus* is $> 10^2$ $TCID_{50}$/ml, although the exact threshold remains to be determined in a more detailed way using a specifically designed experimental protocol. This result is in the line with another mosquito-borne virus model, as a study demonstrated that doses as low as $10^{3.9}$ plaque-forming units (PFU) per ml (approximately $10^4$ $TCID_{50}$/ml) of an Asian genotype Chikungunya virus strain can lead to transmission by *Aedes albopictus* [49]. Viremia levels in phlebovirus infections for human or non-human vertebrates are generally low and transient. The vector requires the ingestion of a significant viral load to enable virus replication and subsequent transmission [13,50–52]. In one of the few experimental studies with *Ph. perniciosus* and TOSV lineage A, the rate of infected sand flies decreased over time despite ingestion of infectious doses greater than $10^6$ $TCID_{50}$/ml [13]. Conversely, our study showed that even when ingesting a medium dose of the virus ($10^4$ $TCID_{50}$/ml), the virus is detectable at least 15 days and the infected sand flies are therefore likely to transmit the virus. This discrepancy between the two studies may be due to (i) the utilization of different TOSV lineages or (ii) the experimental infection methods used. Intra-thoracic inoculations of the virus (far from the natural infection route) might increase sand flies mortality rate over time. After ingestion of a viremic blood meal, the viral load in a competent vector usually increases significantly to reach a plateau which is maintained for the duration of the vector life [24], which is well represented for TOSV as shown in Fig 4. We also detected TOSV RNA in the body of one sand fly at 45 dpi, suggesting that the vector can occasionally remain infected and might be infectious for a longer period. Since sand flies have a longevity of seven to 10 weeks [53], a specific experiment aiming at testing sand flies for a longer period (>45 dpi) would be necessary to determine whether the vector remains infectious. For several arboviruses, the viremia peak of symptomatic human infections is estimated to be around $10^6$ $TCID_{50}$/ml [54,55]. Nevertheless, asymptomatic human cases can achieve viremia levels as high as the lowest estimates for symptomatic cases [56]. To date, no studies have reported the viral load of TOSV in asymptomatic patients. Our results showed that TOSV RNA persists in

sand flies over 15 days when infected at dose as low as $10^4$ TCID$_{50}$/ml. This dose infection threshold appears to be compatible with most viral loads found in reservoirs in other arbovirus models [49]. Therefore, sand flies could be infected throughout the viremic period, regardless of whether the host is symptomatic.

Subsequently, we explored the capacity of sand flies to transmit TOSV, specifically examining their ability to spit out the virus, contained in their saliva, during the feeding process [57]. In colony settings, the blood-fed females died most of the time after egg-laying, before taking a second blood meal [29], precluding to measure transmission during a second artificial blood meal. Therefore, we employed an alternative to this issue, using the vector sugar feeding behavior to detect the virus [58]. To predigest sugar, sand flies release saliva during feeding [59]. We assumed that infected sand flies might be able to spit viral particles in their saliva when they feed on sugar-soaked cotton pads. As, expected, we confirmed that results observed with Massilia virus and *Ph. perniciosus* [18] were similar with TOSV: viral RNA was detected in the sugar cotton pads at two, six, and eight dpi on which sand flies infected with high dose ($10^6$ TCID$_{50}$/ml) had fed. While the presence of TOSV particles at two dpi could be attributed to viral residues (contamination between individuals, defecation, blood residues in mouthparts, etc.), TOSV RNA at six and eight dpi may indicate its presence in the salivary glands. It has already been shown that *Culex spp.* mosquitoes release arboviruses during sugar feeding [60]. The presence of TOSV RNA in the cotton could mean that the virus was disseminated or present in the sand fly salivary glands, but it remains to be determined whether the virus was infectious. Our results suggest that TOSV reaches the salivary glands of *Ph. perniciosus* between three and six dpi although this should be confirmed by fine dissection and testing of only the salivary glands. In addition, the release of TOSV into sugar meal together with recent demonstration of long-term persistence of infectious TOSV in sugar meal (up to 7 days at 26°C) [17] suggest that sand fly to sand fly transmission might occur orally during sap feeding in nature. To better understand the transmission dynamics and be able to predict TOSV emergence it would be crucial to investigate further the possible alternative transmission routes of TOSV between sand flies through their sugar meal.

Transmission becomes possible after the completion of the extrinsic incubation period (EIP), which is considered here as the interval between ingestion of virus and the earliest continued time at which virus is released in the saliva [61]. The average EIP of sandfly-borne phleboviruses is at least one week [62]. The EIP of the phlebovirus Rift Valley Fever virus (RVFV) is a minimum of 14 days after infection [63]. We showed a dose-effect on the systemic infection dynamics, with a median time to reach systemic infection occurring approximately at five and three dpi for sand flies infected by dose 2 ($10^4$ TCID$_{50}$/ml) and dose 3 ($10^6$ TCID$_{50}$/ml), respectively (Fig 4). We can postulate that TOSV median EIPs in *Ph. perniciosus* for each of these doses are at least equal to five and three dpi (Fig 6). The EIP can be influenced by several factors (*e.g.* environmental conditions, vector lifespan or immune system) including the ingested dose of virus [64–66]. In order to accurately estimate the TOSV EIP in the vector, it would be interesting to carry out additional replicates of the sugar cotton pad test and further experiments to examine other parameters that may influence EIP. This parameter is essential since pathogens with a shorter EIP result in earlier transmission after infection [67].

The second objective of our study was to explore the impact of TOSV infection on the sand fly biology. Indeed, effects on vector life-history traits are likely to have a strong impact on virus epidemiology. Initially, we sought to determine if high-dose TOSV infection could influence the vector survival. Our results indicated that the lifespan of infected sand flies was similar to that of uninfected ones (S2 Fig). Therefore, it seems that high-dose TOSV infection have no impact on the vector survival. If the survival of infected sand fly remains unchanged, TOSV can be transmitted throughout the adult vector life (i) during multiple blood meals (for

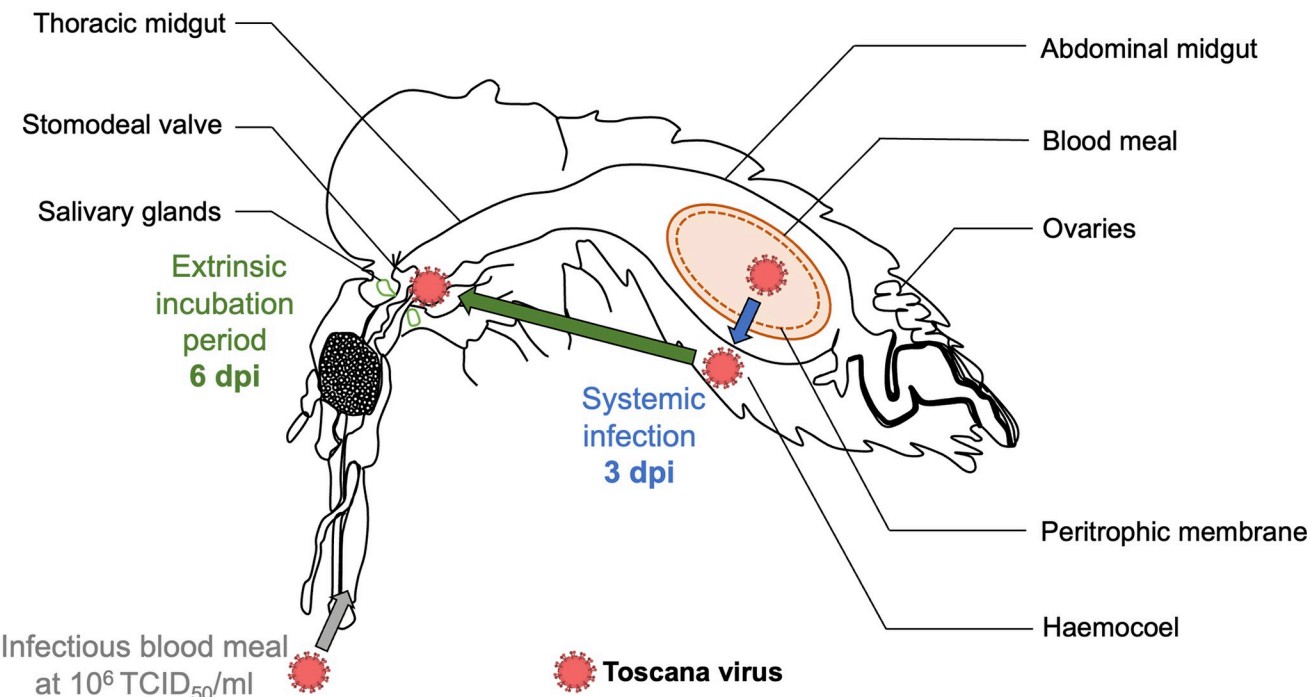

**Fig 6. Schematic of the Toscana virus infection dynamics in the sand fly species *Phlebotomus perniciosus* including estimation of the extrinsic incubation period (green arrow) and the systemic infection (blue arrow) for the highest tested infectious dose $10^6$ TCID$_{50}$/ml.** dpi: days post infection.

females), (ii) during sugar meals (for males and females) and (iii) through multiple oviposition (vertical transmission). Then, we assessed the impact of two TOSV infection doses ($10^4$ and $10^6$ TCID$_{50}$/ml) on various life-history traits of *Ph. perniciosus* to determine the infection impact on vector biology. We showed the impact on the egg hatching time, which was longer for eggs from infected females (Fig 5). As the transovarial transmission of TOSV previously showed in *Ph. perniciosus*, we can anticipate that infected adult sand flies will emerge later, potentially leading to an extension of the period of TOSV intra-species (sand flies) and inter-species (vertebrates and sand flies) transmission [68]. A delay in vector emergence can lead to a reduction in intra- and inter-species competitiveness [69]. We therefore propose an additional hypothesis whereby infected individuals that emerge later may have a better survival probability due to reduced competitiveness. Therefore, the next step should be to monitor the eggs of infected sand flies until adult emergence.

To conclude, we confirmed that *Ph. perniciosus* is a competent vector for TOSV even at a lower dose in the blood meal ($10^4$ TCID$_{50}$/ml) than the virus dose during viremia peak in symptomatic mammals ($10^6$ TCID$_{50}$/ml). All our results suggest that the transmission of TOSV is short (six days) compared to other mosquito-borne viruses (14 days in phlebo-viruses [*e.g.* RVFV] [63], 10 days in flaviviruses [*e.g.* Zika virus] [64]). Given this short transmission, the epidemic risk in TOSV endemic areas must be considered. We have successfully developed and implemented effective protocols for conducting experimental infections with TOSV and *Ph. perniciosus*. We also demonstrated the TOSV infection impact on sand fly life history traits (egg hatching time) that could impact the virus epidemiology in natural populations. These experimental results provide a better understanding of virus maintenance in sand fly populations and the natural cycle of TOSV, while raising new hypotheses on transmission dynamics (Fig 7). Despite providing valuable insights, due to the difficulties in

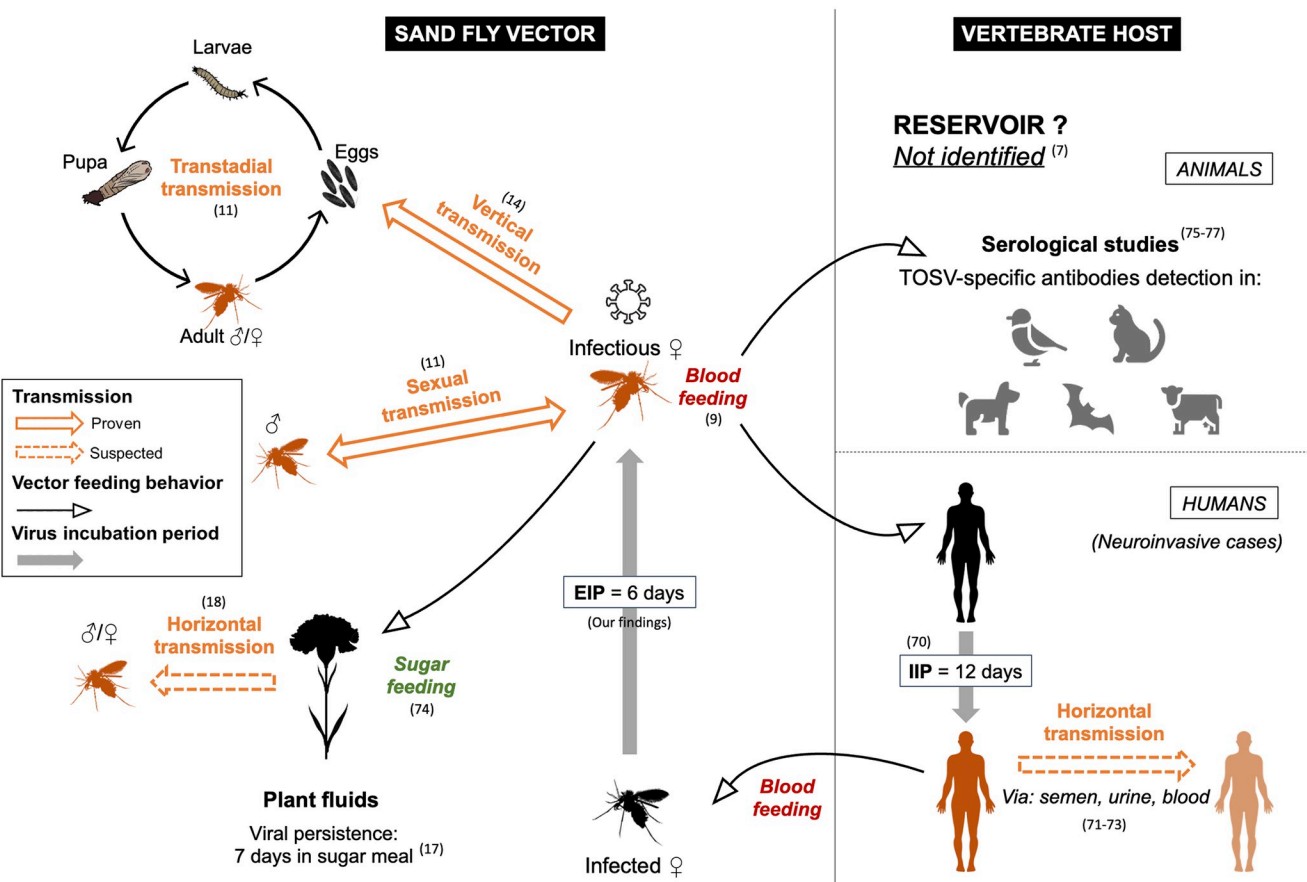

**Fig 7. Toscana virus transmission dynamics within sand fly vectors and vertebrate hosts, including proven and suspected transmission routes, animals identified positive for TOSV-specific antibodies suspected but not identified as reservoirs, and incubation period estimates within the vector (EIP: Extrinsic Incubation Period) and the human neuroinvasive host (IIP: Intrinsic Incubation Period) [7,9,11,14,17,18,70–77].**

rearing sand flies and the long life cycle of this vector, some experiments should be replicated to support the advanced hypotheses. In addition, the estimation of the EIP is based on indirect evidence and requires confirmation through more precise methods, such as virus detection in sand fly salivary glands or stomodeal valve. Finally, vertical transmission of TOSV was not addressed in this study and further work is required to elucidate the existence and impact of this mode of transmission on the persistence of the virus in the environment and its dispersal.

## Supporting information

**S1 Fig. Standard curve for absolute quantification of the Toscana virus RNA by real-time reverse transcriptase-polymerase chain reaction.** All the samples were amplified and considered positive, except samples at $10^{-4}$ RNA copies/ml considered as negative because of late amplification (>36 cycles). The virus detection limit is $10^{-2}$ RNA copies/ml.
(TIF)

**S2 Fig. Kaplan-Meier survival estimates for infected in red (n = 98) and uninfected in green (n = 140) sand flies with Toscana virus with 95% confidence intervals.**
(TIF)

**S3 Fig. Kaplan-Meier hatching time estimates for Toscana virus infected in orange ($10^4$ $TCID_{50}$/ml, n = 62) and red ($10^6$ $TCID_{50}$/ml, n = 62); and uninfected in green (n = 46) sand flies with 95% confidence intervals.**
(TIF)

**S1 Table. Descriptive statistics for the analysis of the infection impact on sand fly life history traits, infected by dose 2 ($10^4$ $TCID_{50}$/ml) and dose 3 ($10^6$ $TCID_{50}$/ml).**
(XLSX)

**S2 Table. Raw data used for the manuscript.**
(XLSX)

## Acknowledgments

Authors thank Dr. Justine Boutry (Tel Aviv University, Tel Aviv, Israel) and Dr. Marc Choisy (OUCRU, Hanoi, Vietnam) for their valuable assistance in the statistical analysis; Justine Fournier, Aloïs Berard (Montpellier University, France) and Dr. Idris Mhaidi (IRD, Montpellier, France) for their help to maintain the sand fly colonies; and Dr. Maarten Schrama (Institute of Environmental Sciences, Leiden University, Leiden, Netherlands) for reviewing the manuscript and providing valuable feedback. A part of the material for viral analyses was provided by the European virus archive-Marseille (EVAM) under the label technological platforms of Aix-Marseille University.

## Author Contributions

**Conceptualization:** Lison Laroche, Anne-Laure Bañuls, Rémi Charrel, Nazli Ayhan, Jorian Prudhomme.

**Data curation:** Lison Laroche.

**Formal analysis:** Lison Laroche, Albin Fontaine.

**Funding acquisition:** Lison Laroche, Anne-Laure Bañuls, Rémi Charrel, Nazli Ayhan, Jorian Prudhomme.

**Investigation:** Lison Laroche.

**Methodology:** Lison Laroche, Anne-Laure Bañuls, Rémi Charrel, Albin Fontaine, Nazli Ayhan, Jorian Prudhomme.

**Project administration:** Lison Laroche, Anne-Laure Bañuls, Jorian Prudhomme.

**Resources:** Lison Laroche, Anne-Laure Bañuls, Rémi Charrel.

**Software:** Lison Laroche.

**Supervision:** Anne-Laure Bañuls, Rémi Charrel, Nazli Ayhan, Jorian Prudhomme.

**Validation:** Lison Laroche, Anne-Laure Bañuls, Rémi Charrel, Albin Fontaine, Nazli Ayhan, Jorian Prudhomme.

**Visualization:** Lison Laroche.

**Writing – original draft:** Lison Laroche.

**Writing – review & editing:** Lison Laroche, Anne-Laure Bañuls, Rémi Charrel, Albin Fontaine, Nazli Ayhan, Jorian Prudhomme.

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
