## [Decision Letter · Decision Letter 0]

9 May 2024

Dear Dr Laroche,

Thank you very much for submitting your manuscript "Sand flies and Toscana virus: intra-vector infection dynamics and impact on Phlebotomus perniciosus life-history traits" for consideration at PLOS Neglected Tropical Diseases. As with all papers reviewed by the journal, your manuscript was reviewed by members of the editorial board and by several independent reviewers. In light of the reviews (below this email), we would like to invite the resubmission of a significantly-revised version that takes into account the reviewers' comments. 

All three reviewers were positive about the manuscript but suggested minor changes. Two of the reviewers have suggested some careful review to make sure conclusions make the data, as well as some very helpful word editing that improves readability. One reviewer had some concerns and questions about the qPCR so be sure to carefully review those.

We cannot make any decision about publication until we have seen the revised manuscript and your response to the reviewers' comments. Your revised manuscript is also likely to be sent to reviewers for further evaluation.

Sincerely,

Brianna R Beechler, Ph.D., DVM

Academic Editor

David Safronetz

Section Editor

All three reviewers were positive about the manuscript but suggested minor changes. Two of the reviewers have suggested some careful review to make sure conclusions make the data, as well as some very helpful word editing that improves readability. One reviewer had some concerns and questions about the qPCR so be sure to carefully review those.

Reviewer's Responses to Questions

**Key Review Criteria Required for Acceptance?**

**Methods**

-Are the objectives of the study clearly articulated with a clear testable hypothesis stated?

-Is the study design appropriate to address the stated objectives?

-Is the population clearly described and appropriate for the hypothesis being tested?

-Is the sample size sufficient to ensure adequate power to address the hypothesis being tested?

-Were correct statistical analysis used to support conclusions?

-Are there concerns about ethical or regulatory requirements being met?

Reviewer #1: The article Sand flies and Toscana virus: intra-vector infection dynamics and impact on

Phlebotomus perniciosus life-history traits investigated the intra-vector TOSV infection dynamics in Ph. perniciosus, as well as its impact on the vector's life history traits. As reported by aithors limited research has been conducted on transmission dynamics and the vectorial competence and capacity of the principal TOSV vector, Phlebotomus perniciosus.

The objectives are clear and the hypothesis stated teastable and the study design appropriate

This manuscript is well written and results presented clearly with only minor formatting/typos that need to be addressed prior to publication:

lines 78-79: please specify species

73-79: please give more details on SF distribution in the mediterranean area especially for Ph. perniciosus

93-103 This paragraph is very broad and general. I suggest summarising it or cutting it.

Reviewer #2: (No Response)

Reviewer #3: A standard curve was carried out for absolute quantification of TOSV RNA in sugar soaked cotton pads through RT-qPCR to obtain the virus detection limit > Was it also done for samples? Or only cottons? 

samples with Ct value < 40 were considered positive > why not using standard curve detection limit value? 

Was TOSV values normalised on sandfly standard gene? To describe. 

To prevent the fungi development which could affect the survival of the sand flies, dead females were removed > how was it performed. I am just concerned if using anesthesia by cold, groups with dead sandflies were therefore more stressed thn other groups so this could biais the outcome on survival. 

Between 10 to 30 individuals were dissected in three parts: head, body (thorax and abdomen), wings and legs, every day for 15 days > It is mentioned L246 that sand flies were dissected every two days. Please correct.

**Results**

-Does the analysis presented match the analysis plan?

-Are the results clearly and completely presented?

-Are the figures (Tables, Images) of sufficient quality for clarity?

Reviewer #1: 117: please explain why you choose Ph. perniciosus

250: Plead explain better in Table 1 that number represent Phlebotomus perniciosus dissected

Figure 3: Add information on the number of doses to the x axis ie: Time post infection (days) (two doses).

Reviewer #2: (No Response)

Reviewer #3: Figure 3. Please do a graph for dose 1. Viral loads are important but on your qPCR graph this is bars so we do not have any idea of the prevalence but just a mean of the whole. Please do a box plot showing all individual values, with qPCR detection limit shown on the graph (dotted line?) and prevalence in % and total number of sandlflies tested visible in figure. 

Why using different colours between dose 2 and 3 on same figure but different panels eg bodies is yellow and then red. Please be homogenous for the ease of readers. Be mindful of colourblind people. 

Legends are a bit short, please add more details, method, numbers, colour code, what the error bars or box plot lines correspond to (sem or sed?) etc. Please refer to published papers to see what a legend should show. 

Increase in viral loads at dose 2. Statistics should be there to support this statement. How were normalised the viral loads, any standard gene? 

In sand flies infected with dose 2, up to 21 days is not shown. Either you do not speak about it or you show it, the number of sandflies at these days is enough according to me, but indeed they come from only one replicate. Maybe best to leave the description of these day out. 

L266 to 270: The analysis of significance of the TOSV dose on post-infection should be shown (table with comparisons and stat). The ext should be more dtailed, which times, which tissue, for instance I can t see any effect of the dose in bodies. The text should be clearer to well describe what s the effect of what and what is significant or not. 

Table 2 should indicate that the values unit is 10 to the 6 (not only in legend). There si a mistake I guess for day 14. 

virus infectivity in sand fly samples: It would be better if there was a figure (in supplementary?) showing on the same graph (two Y axis) both the qPCR and titration data for same samples (the 84 samples) so we can appreciate if viral loads by qPCR is indeed a valid method. And not just a table 2 with only a mean of titres. While qPCR can be a good method to asses viral loads, this needs to be confirmed, especially at the light that none of the legs and heads samples were positive by titration while virus detected by qPCR. 

L309: viral particles were detected > no, RNA was detected. 

A figure should accompany data described L305 to 310. 

infection impact on vector fecundity life history traits > please show a table with all statistics even if not significant. 

only hatching time of test groups dose 2 (GLM test, P = 0.0247) and dose 3 (GLM test, P = 5.61e-07) > does this correspond to p value on figure 5? If yes, the figure should be cited here.

**Conclusions**

-Are the conclusions supported by the data presented?

-Are the limitations of analysis clearly described?

-Do the authors discuss how these data can be helpful to advance our understanding of the topic under study?

-Is public health relevance addressed?

Reviewer #1: Conclusions are well supported by data presented but discussion needs of a specific paragraph on the limitations of the study.

Public health relevance is well adressed

Reviewer #2: (No Response)

Reviewer #3: Authors mention in the abstract that this research confirms vector competence. However, dissemination is not a proof of vector competence. I d recommend being more cautious and say something like potential vector competence. Or change systemic dissemination to presence of virus RNA in saliva/cotton…

L57: on vector hatching time leading to a delayed emergence of infected sand flies > check wording here so this is correct. Larval hatching and delayed emergence of corresponding adults (not infected but their mothers were), modify the sentence. Actually in methods it is mentioned that alrvae were killed so what is delayed emergence? Eggs hatched into larvae and adults emerge from pupae. Use the correct vocabulary to avoid any confusion. 

As mentioned in general comments, while it is not highly problematic to use qPCR, conclusions and discussion must stick to the data and authors should not overstate. The results as they are are interesting and constitute a good advance in the field of sandfly vector competence. 

Maybe try to make the point that finding RNA copies in head or in cotton means that there has been dissemination or presence of virus in salivary glands (RNA of dead particles could be spitted?) but stay cautious as it does not mean it is still infectious or that the saliva was. Just stay on the cautious side and highlight the limits of the study. 

The authors should carefully revise their manuscript, so their conclusions fit to their data and not to overstate. The limitations of RTqPCR should be highlighted and discussed (see figure suggested to compare qPCR and titrations on samples measured both ways)

Examples where authors should revise (not exhaustive, please carefully check all statements in discussion as I do not list them all). 

“we found viral particles in head and thorax, where the salivary glands are presents (42), until at least 15 dpi, as well as in wings and legs (Fig 3)”. On fig 3 these are RNA not viral particles. 

“virus particles did not disseminate in the vector, and only a few body samples were tested positive before sand fly defecation”

“TOSV disseminated into the secondary tissues and the infection persisted for at least 15 days (Fig 4)”, only tested for dose 3 (infection means infectious virus particles, not RNA which can be traces of dead particles). 

L 402, L423 (sugar transmission, watch as you show RNA here) Etc

L460 to463: it is a bit of a stretch here. Transovarial transmission does not mean necessarily adult females infected. You also never show that you have transovarial transmission in your model. There is also a lack of clarity between hatching and emergence. They never tested emergence, so stick to hatching. Rather than over hypothesising on adult mergence, I d rather try to discuss why hatching is longer, what would be the underlying mechanism (reproductive process less effective in inf females, leading to less reserves?)

I am not too sure we can compare EIP in sandflies and mosquitoes. In addition, some arboviruses transmitted by mosquitoes have a very short EIP eg CHIKV. 

The discussion is generally a bit too long, but I guess this can be shortened by removing all overstatement and focusing on the data without stretching on hypotheses.

**Editorial and Data Presentation Modifications?**

Reviewer #1: (No Response)

Reviewer #2: (No Response)

Reviewer #3: It would be good to mention in introduction, if known, the approx. numbers or %age of meningitis and encephalitis. 

L32 : the virus's > the viral

L34: vectorial competence and capacity > vector competence and vectorial capacity

L37: though> through

L38: potentially resulting in a shorter extrinsic incubation period > shorter than what?

L41: TOSV’s > TOSV. In general, check all the possessive forms as this is overused at different places in the manuscript

L63: space between tri and segmented to remove

L138: of 0.1 after five days of post infection> and supernatant collected after five days? Clarification of supernatant? 

L144: infection> infections 

The three doses are called sometimes small, medium and large dose, and sometimes 1,2,3. Find one notation and keep it through the manuscript. Actually it would be easier to write the dose each time (text and figure legends) instead of 1, 2, 3 or small etc

L181: the traits most important in the virus transmission > other traits are important so rephrase eg vector survival and fecundity, two important traits…

L287: tree > three

L324: eggs and larvae numbers, oviposition and hatching times> please invert so respect the chronology

L336: standard error > sem or sd? 

L344: p-value significance: 0 '***' > something missing here, cannot be 0 but 0.0001? 

L354: Vectorial competence > vector competence and vectorial capacity . Please check all the manuscript

L361: vector competence is the ability to be infected and further transmit ( not just transmit)

L 364: where it may be released in saliva during a subsequent blood meal > released in saliva and transmitted during a subsequent blood meal

L 373: only a few body samples were tested positive before sand fly defecation> can you clarify why mentioning defecation here? What s the significance? Btw, prevalence must be shown for such statements. 

L380: with other arbovirus model> with another… or with …models

L384: are generally low and transient, often necessitating the ingestion of a substantial viral load by the vectors > not sure what is the causal link here, not clear. Please reformulate. 

L408: female sand flies die before taking a second blood meal > I guess this depends when you offer them a second blood meal.. please clarify, is this that they die before saliva is infectious??? It does not makes sense as you manage to get RNA on cottons from still alive females. 

L460: As the transovarial transmission of TOSV exists in Ph. Perniciosus > add ref

**Summary and General Comments**

Reviewer #1: This manuscript is well written and results presented clearly .

Reviewer #2: (No Response)

Reviewer #3: In this study, Laroche at al. analyse TOSV infection dynamics in the sandfly Phlebotomus perniciosus, a main vector of TOSV. They also analyse the impact of infection on different fitness traits of the vector, survival, fertility and fecundity. This is quite an impressive study, considering the high complexity of working and breeding sandflies, with replicates and high number of flies. Although some conclusions can be made around infectivity, dissemination and transmission, most of the data were obtained by qPCR which does not measure infectious viral particles. While I appreciate the complexity of such experiments in a BSL3 lab, and that the results are valid experimentally, I think that some conclusions are a bit overstated, ie a RNA copy does not prove infectivity. Saying this, data obtained by qPCR are still very interesting and authors can pull some conclusions from it but they should revise the manuscript not to overstate and highlight the limitations. See my other comments in other sections about this. 

Overall, the manuscript is well written, methods and results well described. I suggested some minor modifications and some changes in result presentation, including a couple more figures and more description in legends. 

Considering that this study is one of the very few in this field (sandflies and arboviruses), this manuscript is really of interest and merits publication in Plos NTD. However, the authors should carefully revise their manuscript according to my comments to improve its quality and that readers are aware of some limitations. I clicked major revision, but they are all text edits.

PLOS authors have the option to publish the peer review history of their article (what does this mean?). If published, this will include your full peer review and any attached files.

Reviewer #1: No

Reviewer #2: No

Reviewer #3: No
---

## [Decision Letter · Decision Letter 1]

4 Sep 2024

Dear Dr Laroche,

We are pleased to inform you that your manuscript 'Sand flies and Toscana virus: intra-vector infection dynamics and impact on Phlebotomus perniciosus life-history traits' has been provisionally accepted for publication in PLOS Neglected Tropical Diseases.

Best regards,

Brianna R Beechler, Ph.D., DVM

Academic Editor

David Safronetz

Section Editor

Reviewer's Responses to Questions

**Key Review Criteria Required for Acceptance?**

**Methods**

-Are the objectives of the study clearly articulated with a clear testable hypothesis stated?

-Is the study design appropriate to address the stated objectives?

-Is the population clearly described and appropriate for the hypothesis being tested?

-Is the sample size sufficient to ensure adequate power to address the hypothesis being tested?

-Were correct statistical analysis used to support conclusions?

-Are there concerns about ethical or regulatory requirements being met?

Reviewer #2: (No Response)

Reviewer #3: (No Response)

**Results**

-Does the analysis presented match the analysis plan?

-Are the results clearly and completely presented?

-Are the figures (Tables, Images) of sufficient quality for clarity?

Reviewer #2: (No Response)

Reviewer #3: (No Response)

**Conclusions**

-Are the conclusions supported by the data presented?

-Are the limitations of analysis clearly described?

-Do the authors discuss how these data can be helpful to advance our understanding of the topic under study?

-Is public health relevance addressed?

Reviewer #2: (No Response)

Reviewer #3: (No Response)

**Editorial and Data Presentation Modifications?**

Reviewer #2: (No Response)

Reviewer #3: (No Response)

**Summary and General Comments**

Reviewer #2: (No Response)

Reviewer #3: I am happy with the revisions.

PLOS authors have the option to publish the peer review history of their article (what does this mean?). If published, this will include your full peer review and any attached files.

Reviewer #2: No

Reviewer #3: No

---

## [Editor Report · Acceptance letter]

20 Sep 2024

Dear Dr Laroche,

We are delighted to inform you that your manuscript, "Sand flies and Toscana virus: intra-vector infection dynamics and impact on Phlebotomus perniciosus life-history traits," has been formally accepted for publication in PLOS Neglected Tropical Diseases.

Best regards,

Shaden Kamhawi

co-Editor-in-Chief

Paul Brindley

co-Editor-in-Chief
